# Steering and cloaking of hyperbolic polaritons at deep-subwavelength scales

Hanchao Teng[1,2,6], Na Chen[1,2,6], Hai Hu [1,2]✉, F. Javier García de Abajo [3,4] & Qing Dai [1,2,5]✉

Polaritons are well-established *carriers* of light, electrical signals, and even heat at the nanoscale in the setting of on-chip devices. However, the goal of achieving practical polaritonic manipulation over small distances deeply below the light diffraction limit remains elusive. Here, we implement nanoscale polaritonic in-plane steering and cloaking in a low-loss atomically layered van der Waals (vdW) insulator, α-MoO₃, comprising building blocks of customizable stacked and assembled structures. Each block contributes specific characteristics that allow us to steer polaritons along the desired trajectories. Our results introduce a natural materials-based approach for the comprehensive manipulation of nanoscale optical fields, advancing research in the vdW polaritonics domain and on-chip nanophotonic circuits.

The pursuit of light propagation at extreme subwavelength scales has been a prominent subject within nanophotonics. Achieving control over this phenomenon is pivotal for the realization of photonic circuits and on-chip devices[1,2]. In this context, metamaterials offer a constructive paradigm for manipulating light through the arrangement of numerous subwavelength unit cells[3–6]. In addition to artificial structures, polaritons in natural materials, which are hybrid light-matter modes, offer a powerful framework for light control with field confinement far below the diffraction limit[7–14]. Thus, polaritons have emerged as effective carriers of light, electrical signals, and even heat at the nanoscale within on-chip circuits[15].

The behavior of polaritons depends on intrinsic material properties. Hyperbolic materials possess permittivity components having opposite signs along different crystal directions, which leads to the creation of hyperbolic dispersion contours for polaritons. In recent years, hyperbolic polaritons in vdW materials have attracted much attention due to their high spatial confinement, long lifetimes, and extreme anisotropy[16–22]. Multiple versions of hyperbolic polariton modes[23–25] and anomalous propagation characteristics[26–37] have been demonstrated as well. These rapid advancements in polaritonic research of appealing materials have opened up exciting possibilities

for achieving practical polaritonic steering at the nanoscale. Such a function requires the integration of complex transmission characteristics across different interfaces, which has remained elusive thus far.

In this study, we introduce a strategy that enables the steering and cloaking of hyperbolic polaritons, which is leveraged by vdW crystals of α-MoO₃ as the fundamental building blocks of customizable stacked and assembled structures with great versatility. Each block contributes layer-specific characteristics that effectively mold the flow of polaritons along desired trajectories. Based on a high polaritonic transmission across various structures and interfaces, which benefits from the robust hybridization and strong modal-profile alignment of phonon polaritons, we demonstrate in-plane polaritonic cloaking devices deeply below the light diffraction limit. Our study provides a promising platform for realizing practical polaritonic circuits.

## Results

### Directional control of transmission by twist angles

In our experiments, hyperbolic polaritons in the bottom α-MoO₃ film are launched by a resonant gold antenna[38], as depicted in Fig. 1a and detailed in Methods and Supplementary Note 1. α-MoO₃ is a highly anisotropic vdW material that supports hyperbolic polaritons along

[1]CAS Key Laboratory of Nanophotonic Materials and Devices, CAS Key Laboratory of Standardization and Measurement for Nanotechnology, National Center for Nanoscience and Technology, Beijing 100190, P. R. China. [2]Center of Materials Science and Optoelectronics Engineering, University of Chinese Academy of Sciences, Beijing 100049, P. R. China. [3]ICFO-Institut de Ciencies Fotoniques, The Barcelona Institute of Science and Technology, Castelldefels (Barcelona) 08860, Spain. [4]ICREA-Institució Catalana de Recerca i Estudis Avançats, Passeig Lluís Companys 23, 08010 Barcelona, Spain. [5]School of Materials Science and Engineering, Shanghai Jiao Tong University, Shanghai 200240, P. R. China. [6]These authors contributed equally: Hanchao Teng, Na Chen. ✉e-mail: huh@nanoctr.cn; daiq@nanoctr.cn

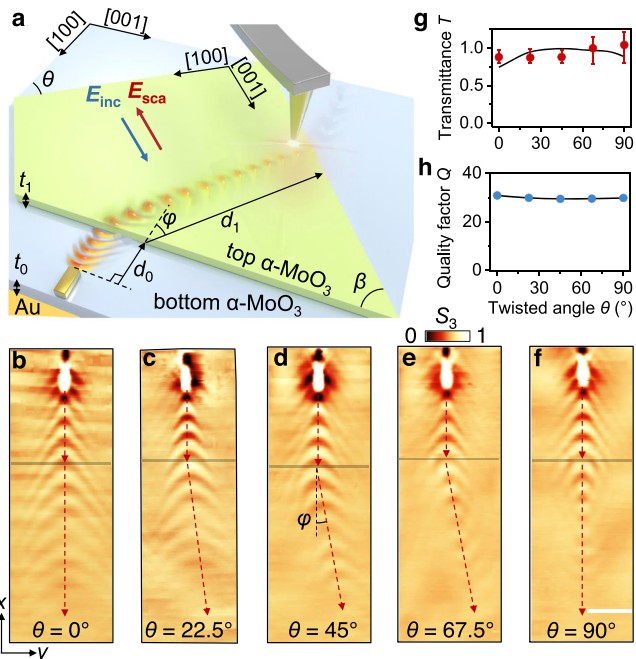

**Fig. 1 | Refractive transmission of hyperbolic polaritons. a** Schematic of the device structure and experimental setup consisting of two twisted α-MoO$_3$ films, a launching gold antenna on the bottom one, and a probing near-field tip hovering the structures. $E_{inc}$ and $E_{sca}$ represent the incident and scattering electromagnetic waves, respectively. $\theta$ indicates the twist angle between the in-plane crystallographic orientation of the two films; $\beta$ represents the angle of the cut edge relative to the crystal axis [001] of the top α-MoO$_3$ film; and $\varphi$ indicates the deflection angle of refracted polaritons (at the edge delimiting the boundary between the single film and the double film) relative to the incident ones. The cut edge (i.e., the interface) of the top film is kept perpendicular to the antenna long axis in our experiments, such that polaritons impinge at normal incidence. The thickness of the α-MoO$_3$ films at the bottom and top is denoted as $t_0$ and $t_1$, respectively. $d_0$ and $d_1$ represent the propagation distance of polaritons in the bottom and twisted α-MoO$_3$ region, respectively. **b–f** Experimental near-field images (amplitude signal $S_3$) recorded at different twisted angles $\theta = 0°$, 22.5°, 45°, 67.5°, and 90°. The thicknesses of the bottom and top α-MoO$_3$ films are $t_0 = 550$ nm and $t_1 = 150$ nm, respectively. The illumination frequency is fixed at 893 cm$^{-1}$. The gold antenna is positioned $d_0 \approx 5$ μm away from the interface (represented by horizontal gray lines). Red-dashed arrows indicate the propagation direction of polaritons. The scale bar indicates 3 μm. Note that experimental near-field images are normalized in this work, as well as the simulated images. **g** Experimentally measured (red dots) and numerically simulated (black curve) polariton transmittance across the interface for various twist angles $\theta$. Error bars indicate 95% confidence intervals. **h** Ad hoc overall quality factor of the mixed (hyperbolic and hybrid) polaritons in the stacking structures as a function of twist angle.

the [100] crystal direction within the Reststrahlen band II (816 cm$^{-1}$–972 cm$^{-1}$)$^{19,39}$, which we capitalize in this work. These polaritons refract at the interface of the twisted double films, propagate inside the twisted region, and refract again into the original film. Figure 1a sketches real-space infrared nanoimaging of polariton propagation and transmission in this geometry using scattering-type scanning near-field optical microscopy (s-SNOM) (Supplementary Figs. 1 and 2). The measured images in Fig. 1b–f illustrate polariton refraction at different twist angles. The regions above the gray horizontal line represent the bottom single α-MoO$_3$ film, where polaritons propagate with regular hyperbolic wavefront. However, a topological transition occurs due to the robust hybridization of hyperbolic polaritons in the twisted double films$^{26,27,40,41}$ (see the theoretical model in Supplementary Fig. 3 and Supplementary Note 2 along with the corresponding isofrequency contours (IFCs) in Supplementary Fig. 4). We have further marked the refraction process of various typical

incident directions. By conserving the polariton wave vector along the direction of the refracting interface, we can obtain the corresponding refraction direction for each incident wave vector (Supplementary Fig. 4). Upon transmission of hyperbolic polaritons across the interface, our analysis reveals a transition from normal to negative refraction by rotating the top α-MoO$_3$ film.

The in-plane symmetry of the polariton IFC is broken when the interface is not aligned with the natural crystal axes of α-MoO$_3$, leading to a change in the direction of polariton refraction, as shown in Fig. 1c-e. As the twist angle $\theta$ increases, the direction of polariton propagation gradually deviates from the original direction after crossing the interface. The deflection angle $\varphi$ is dependent on the orientation of the polaritonic IFCs in the twisted region (Supplementary Fig. 5). Besides, the dielectric function in twisted α-MoO$_3$ can be expressed as a linear superposition of the dielectric functions of the layers$^{42}$, which results in the presence of off-diagonal terms. Therefore, the distorted and asymmetrical hyperbolic modes in the twisted region can be interpreted as a shear mode that has recently been observed in natural crystals with low symmetry such as monoclinic $\beta$-Ga$_2$O$_3$ and CdWO$_4$$^{24,25}$.

To quantify losses during propagation and refraction, we extract the quality factor $Q$ associated with the overall propagation path including transmittance at the interfaces, and defined in terms of the polariton intensities in different traversed spatial regions (Supplementary Fig. 6 and Supplementary Note 3). Thanks to the inherent robustness of the topological transition and the appropriate match of modal profiles, the transmittance remains over 85% to 95% at various twist angles, as demonstrated in Fig. 1g and Supplementary Note 4 and Figs. 7 and 8. The trend of the change in the experimentally measured transmittance with the twisting angle does not match well with simulations. We attribute this disagreement to the fact that negative refraction produces focusing and enhances the transmittance$^{37}$.

Compared to conventional bulk material, the layered structure formed by vdW interactions minimizes the introduction of structural roughness during fabrication, leading to a reduced level of scattering losses at the interfaces. Besides, gold flakes serve as a flat low-loss substrate in our experiments because a new image mode is formed, which stems from the coupling between collective charge oscillations and hybridization of polaritons with their mirror image in the metal. Notably, the image phonon polaritons provide both stronger field confinement and a longer lifetime compared to phonon polaritons on a dielectric substrate$^{43-47}$. Therefore, having such low interface losses, combined with the near-field amplitude enhancement produced by the mirror image in the gold substrate, the ad hoc overall quality factors remain around 30 for different twist angles (Fig. 1h). The experimentally observed propagation direction and fringe wavelength (Fig. 1b-f) are in agreement with numerical simulations (Supplementary Fig. 9).

## Steering polaritons with differently oriented microribbons

The long propagation and low refraction losses of mixed polaritons in twisted α-MoO$_3$ render these structures a powerful platform for demonstrating advanced optical functionalities. We accomplish this potential by assembling highly anisotropic α-MoO$_3$ ribbons with various cut orientations, for which we choose the ribbon width close to the polariton wavelength. As depicted in Fig. 2a and Supplementary Figs. 10 and 11, nanofabrication with typical exposure and etching conditions enables cutting α-MoO$_3$ microribbons with the required widths and smooth edge quality.

When the hyperbolic polariton is launched in the underlying α-MoO$_3$ film (Fig. 2b, c, h, i), it undergoes refraction twice as it traverses the region decorated with each microribbon (Fig. 2d, e, j, k). This results in a horizontal deflection of the propagation direction (we accompany the near-field images with red-dashed arrows indicating Poynting vector propagation directions), performing a zigzag-shaped

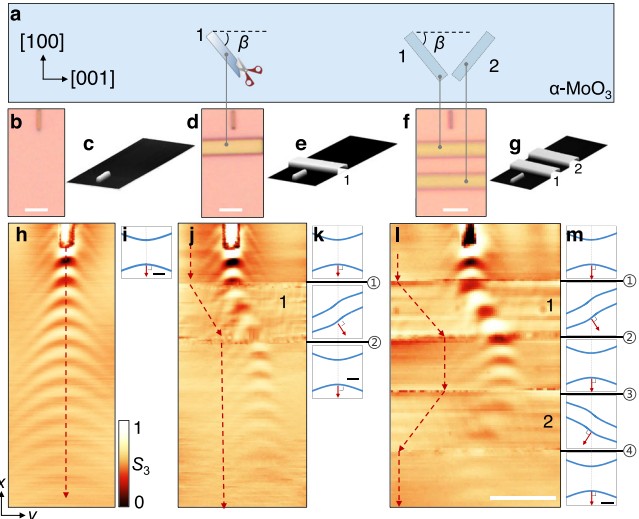

**Fig. 2 | In-plane steering of polaritons with misaligned crystallographic orientations. a** Illustration of tailored α-MoO₃ microribbons with different cut angles $\beta$ relative to the crystallographic orientation of a common source film. **b**–**g** Optical images (**b, d, f**) and atomic force microscopy (AFM) images (**c, e, g**) of different polaritonic devices composed of a bottom α-MoO₃ film and tailored α-MoO₃ microribbons. Two α-MoO₃ microribbons with $\beta = 45°$ (labeled 1) and $\beta = 135°$ (labeled 2) are used in (**b**–**g**). **h, j, l** Near-field amplitude images corresponding to the devices in (**b, d, f**), respectively. The polariton propagation path is controlled by the top tailored α-MoO₃ microribbons: one deflection at the microribbon labeled 1 in (j), leading to a lateral shift of polaritons; and two deflections with opposite angles in (l), leading to a final undeflected transmitted beam. Red-dashed arrows indicate the polariton propagation direction dictated by the Poynting vector **S**, as obtained from the IFCs analysis presented in (**i, k, m**). The thicknesses of the bottom film and top ribbons are $t_0 = 184$ nm and $t_1 = 154$ nm in (h, j, l). Scale bars in (**b, d, f, h, j, l**) indicate 3 μm. **i, k, m** Calculated isofrequency contours of polaritons (blue curves) corresponding to each region in the devices. Horizontal black lines (labeled ①-④) indicate interfaces between different regions. Red arrows represent Poynting vectors **S**, directed along the energy flow and normal to the IFCs. Scale bars indicate 20 $k_0$, where $k_0$ indicates the incident wavevector.

waveguided steering similar to those found in photonic crystals[48]. The analysis of IFCs allows for a precise determination of the propagation direction of polariton energy flow. Note that the complexity of the material anisotropies renders Poynting vectors with a broad range of components, but we only mark the primary direction.

Although the polaritons cross the interface twice, optical losses are surprisingly low, as evidenced by the fact that the fringe intensities in Fig. 2h and j remain essentially constant. Therefore, we can more aggressively install a second transverse ribbon in the propagation direction of the polariton. When the two adjacent ribbons have identical cutting-edge angles, the direction of propagation shifts in the same direction twice (Supplementary Fig. 12). Instead, when the edge angles are mirror-symmetric, the second refraction causes the propagation direction to return to the initial incidence direction (Fig. 2f, g, l, m). Supplementary Fig. 13 compares the experimental and simulated refractions and illustrates the polariton coupling process in different structures through the cross-sectional electric field. High wave vector modes may be lost at the interface because the asymmetry of IFCs makes it challenging to satisfy all wave vector matches of the two sides in the fabricated structures.

### Refraction-based polariton cloaking
To realize a more sophisticated optical functionality, we fabricated an in-plane cloaking device. Previously, this has been accomplished by employing special designs of materials or structures to refractively obscure an object[49,50] or diminish its scattering strength toward light

signals for cloaking carpets of multiple colors and broad bands[51,52]. Here, we build a polariton cloaking device made up of four microribbons that are symmetrically arranged with two different orientations (Fig. 3a, b). This is enabled by an AFM-probe-based transfer method that allows us to accurately move the α-MoO₃ microribbons (Supplementary Fig. 10).

The hyperbolic waves are split and deflected by these ribbons, with each of the two split beams undergoing two deflections such that a central region in the structure is hidden from the polaritons, as shown in the experimental near-field image in Fig. 3c. Increasing the cut-edge angle, width, and thickness of the ribbons should extend the cloaking region in the $y$ direction. We have conducted a thorough comparison of polariton transmission and phase accumulation in this device before and after introducing a defect (a graphite disk) to assess the effectiveness of the cloaking effect (Supplementary Fig. 14). The extracted near-field amplitude profiles demonstrate that the defect has little impact on the intensity and wavelength of the polariton wave (Fig. 3d,e). The electromagnetic simulations provide further evidence of the complex field distribution and involved propagation path in the cloaking device (Fig. 3f), in good agreement with experimental observations (Fig. 3c). In contrast, the presence of a defect along the propagation path of the polaritons without a cloaking structure induces a substantial reduction in the intensity (Supplementary Fig. 15). This effect is primarily attributed to the scattering and reflection of propagating polaritons at the defect edge as well as the modified dielectric environment within the defect region.

We note that most of the optical cloaking works in the literature have so far been realized through transformation optics[53,54]. This approach has revolutionized the field of cloaking with a great deal of freedom, flexibility, and high precision in designing and implementing cloaking devices. We employ anisotropic refraction to steer polariton propagation and enable cloaking, which arises from a topological transition in the iso-frequency contours due to the hybridization of different hyperbolic modes. Therefore, our study may provide a foundation for future designs as well as validation of transformative polaritons involving the use of pseudo-continuous media or metamaterials to confine these excitations.

## Discussion
We demonstrate the in-plane steering and cloaking of strongly confined hyperbolic polaritons using carefully designed double-layer vdW heterostructures through in-plane stacking and splicing of α-MoO₃ with various orientations. This approach allows us to steer polaritons along any desired trajectories, leading to the demonstration of in-plane cloaking devices at deep subwavelength scales. Customizable stacked and spliced structures of natural vdW materials provide high quality factors and low interface losses. From a scientific perspective, our findings open up a wealth of possibilities for advances in transformation polaritonics and represent a solid step in the quest towards achieving the ultimate optical manipulation goal through a meticulous organization of atomically thin interfaces[55,56]. Technologically, our work has substantial potential for the development of nanoscale optical circuits and devices.

## Methods
### Device nanofabrication
High-quality α-MoO₃ and graphite bulk crystals were synthesized using the chemical vapor deposition (CVD) method and mechanically exfoliated onto SiO₂ (300 nm)/Si (500 μm) substrates (Shanghai Onway Technology Co., Ltd). Optical microscopy was used to select suitable α-MoO₃ films, whose thicknesses were subsequently determined using atomic force microscopy (AFM).

A 100 kV electron-beam lithography (EBL) setup (Vistec 5000 + ES, Germany) was used to define patterns with different cut angles of α-MoO₃ on ~1 μm thickness of PMMA950K lithography resist

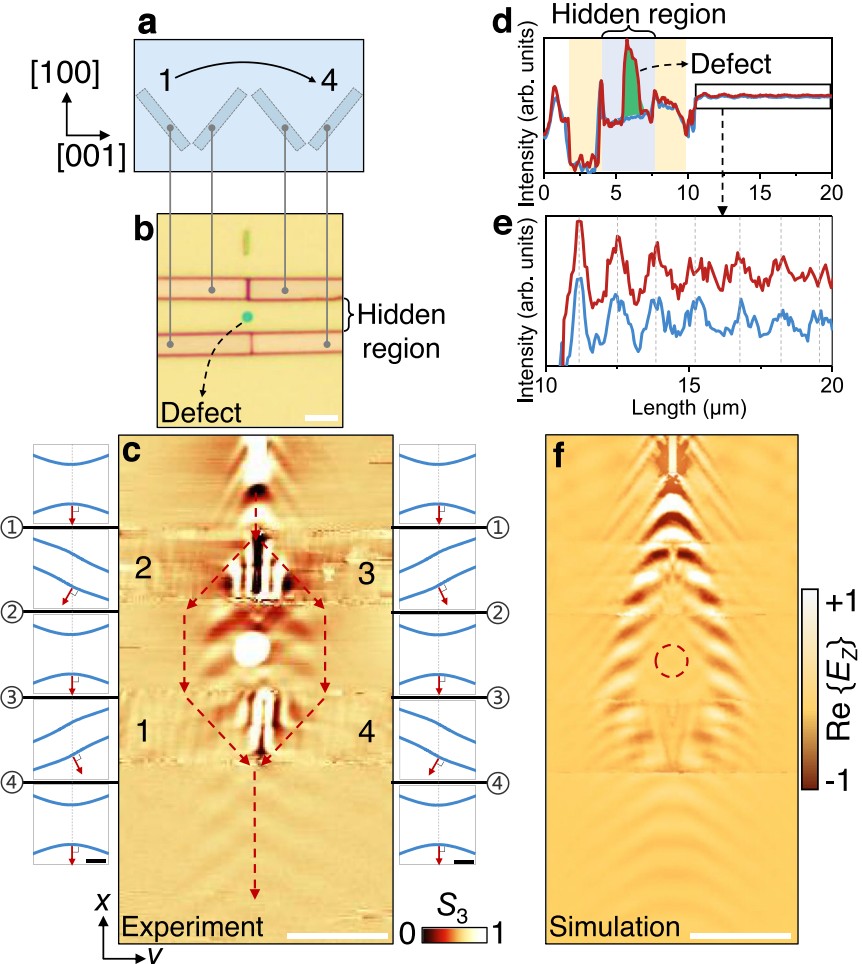

**Fig. 3 | Hyperbolic polariton cloaking. a** Illustration of the crystallographic orientation of α-MoO$_3$ microribbons used in the cloaking device and tailored from the same film. **b** Optical image of a polaritonic cloaking device composed of four microribbons with $\beta = 45°$ (ribbons 1 and 3) and $\beta = 135°$ (2 and 4). The thicknesses of the bottom film and four top ribbons are $t_0 = 207$ nm and $t_1 = 143$ nm, respectively. The green dot is a graphite disk (50 nm thickness, 1 μm diameter), which serves as a cloaked defect. **c** Experimentally measured near-field amplitude images from the device in (**b**) at an illumination frequency of 900 cm$^{-1}$. The incident hyperbolic wave undergoes splitting and subsequent recombination, thus realizing in-plane cloaking of the graphite defect. Red-dashed arrows indicate the polariton propagation direction dictated by the Poynting vector, as obtained from the IFCs

analysis presented on both sides of the experimental image. The calculated IFCs for each region in the device are shown as blue curves in the left and right parts, with scale bars indicating 20 $k_0$. **d** Measured near-field profiles of the cloaking device with (red) and without (blue) the defect placed in the hidden region (blue shaded area). The green shaded area depicts the near-field intensity of the defect. The data is extracted along the red and blue dashed vertical lines in Supplementary Fig. 11. **e** Close-up view of near-field profiles in (**d**). Gray vertical dashed lines represent the position of each peak of the near-field profiles. **f** Simulated near-field (Re{$E_z$}) image illustrating the cloaking performance. The red-dashed circle marks the location of the defect in correspondence with the experimental structure. Scale bars in (**b**, **c**, **f**) indicate 3 μm.

(RDMICRO Inc.). The patterns were etched with SF$_6$ and Ar using reactive ion etching (RIE). The samples were further treated by immersion in hot acetone at 80 °C for 20 min and IPA for 3 min to remove any residual organic materials, followed by nitrogen gas drying.

To construct the gradually rotatable α-MoO$_3$ structure in the main text, we used a deterministic dry transfer process with a PDMS/PC stamp. First, the mechanically exfoliated α-MoO$_3$ flakes were transferred onto gold (60 nm)/Si (500 μm) substrates. Then, the α-MoO$_3$ patterns were transferred onto the α-MoO$_3$ flakes step by step (*i.e.,* one at a time). Specifically, we used a plateau AFM tip to push or rotate the structures to specific positions.

Gold antenna arrays were patterned on the devices using approximately 350 nm of PMMA950K lithography resist. We deposited 50 nm of Au using electron-beam evaporation in a vacuum chamber at a pressure of <5×10$^{-6}$ Torr, followed by liftoff to remove any residual organic materials and the Au film.

In the cloaking device, we also used a graphite disk with 50 nm thickness and 1 μm diameter as a defect for the following reasons: firstly, this structure is easy to process and allows for precise manipulation; secondly, it introduces significant interference in the transmission of polaritons; and thirdly, it avoids excessive excitation of polaritons, minimizing interference with experimental observations.

## Scattering-type scanning near-field optical microscopy (s-SNOM) measurements

**s-SNOM measurements.** We utilized a commercially available s-SNOM (Neaspec GmbH) to perform infrared nanoimaging of polaritons in α-MoO$_3$. The system employed a platinum-coated atomic force microscope tip (NanoWorld) with an approximate radius of 25 nm as the primary scanning platform for approaching and scanning the sample. A monochromatic mid-infrared light source from a quantum cascade laser (QCL) with a tunable frequency range of 890 to 2000

cm$^{-1}$ was used to illuminate the tip. The laser beam, with p-polarization and a lateral spot size of around 25 μm, was focused through a parabolic mirror at an incident angle of 55° to 65°. This setup effectively covered a large area of interest in the samples. The near-field nanoimages were captured by a pseudoheterodyne interferometric detection module, with the AFM tip-tapping frequency and amplitudes set to approximately 270 kHz and 30-50 nm, respectively. The detected signal was demodulated at the third harmonic (denoted $\mathbf{S}_3$) of the tapping frequency to obtain near-field amplitude images that were free of any background interference.

**Polariton launched by gold antenna.** In s-SNOM measurements, polaritons can be excited by a variety of structures, including tips, antennas, edges, and even defects. We primarily utilize metal antennas as the excitation source due to their ability to separate the excitation and detection processes. This separation enables us to directly observe a diversity of refractive transmission of polaritons. When tip excitation is employed, only the interference fringes of the mode would be observed, and the refractive transmission cannot be directly visualized.

In addition, metal antennas can provide high excitation efficiency. When infrared light irradiates the metal antenna, it can excite the plasmon resonance in the antenna and form an in-plane oscillating dipole, thereby exciting the polaritons in the sample with high efficiency. To do this, we designed resonant antennas with a length of approximately 3.0 μm.

## Data availability
The data that support the findings of this study are available within the paper and the Supplementary Information. All raw data generated during the current study are available from the corresponding authors upon request.

## Code availability
All codes generated during the current study are available from the corresponding authors upon request.

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

## Acknowledgements

The authors acknowledge Prof. D. N. Basov (Columbia University), Prof. Cheng-Wei Qiu and Dr. Tan Zhang (National University of Singapore) for valuable discussions and are grateful to Dr. Xiaoyu Li at Institutional Center for Shared Technologies and Facilities of Institute of Process Engineering, Chinese Academy of Sciences for the help in taking and processing FIB-SEM characterizations. This work was supported by the National Natural Science Foundation of China (Grant Nos. 52322209, 52350314, 52172139 to H.H., and 51925203, U2032206 to Q.D.), National Key Research and Development Program of China (Grant Nos. 2021YFA1201500 to Q.D. and 2020YFB2205701 to H.H.), Beijing Nova Program (Grant No. 2022012 to H.H.), Youth innovation promotion association of Chinese Academy of Sciences (Grant No. 2022037 to H.H.); and Strategic Priority Research Program of Chinese Academy of Sciences (Grant No. XDB36000000 to Q.D.); F.J.G.d.A. acknowledges the ERC (Grant No. 789104-eNANO), and the Spanish MICINN (Grants Nos. PID2020-112625GB-I00 and SEV2015-0522).

## Author contributions

Q.D. and H.H. conceived the idea. Q.D. and F.J.G.A. supervised the project. H.H. and N.C. led the experiments, prepared the samples, and performed the near-field measurements. H.T. developed the theory and performed the simulation. H.H., H.T., N.C., Q.D. and F.J.G.A. analyzed the data and discussed the results. H.H., N.C., H.T. and Q.D. co-wrote the manuscript, with input and comments from F.J.G.A.

## Competing interests

The authors declare no competing interests.
