## [Peer Review File · Nature Communications]

Steering and cloaking of hyperbolic polaritons at deep-subwavelength scalesEditorial Note: Parts of this Peer Review File have been redacted as indicated to maintain the confidentiality of unpublished data, and to remove third-party material where no permission to publish could be obtained.

REVIEWER COMMENTS

Reviewer #1 (Remarks to the Author):

In this manuscript, Teng et al constructed interfaces between a in-plane hyperbolic MoO₃ and its twisted double layer, in which in-plane isofrequency curves can be controlled by the twist angle. In this manner, one can control the PhP propagation across the interface and achieve a plenty of noble optical phenomena. Specifically, I believe that planar refraction behavior of hyperbolic polaritons is worthy of study. However, I think the results demonstrated in the current study are still not enough convincing to me due to the following aspects.

1. When analysis the PhP propagation across the interface between single-layer and double-layer MoO₃, it looks only the waves with momentum along the x direction are considered. How about the PhP with off-axis k ? Did they also refract at the interface and how are their refraction?
2. According to the imaging principle of s-SNOM, a collected near-field amplitude is the sum of the antenna-launched PhPs waves and the local scattering amplitude results from the local dielectric response of the material beneath the tip. Intuitively, the local scattering amplitude at the double-layer should be higher than that at the single-layer, therefore, an amplitude abrupt is expected at the interface between the single- and double-layer region. In Figure S4, the amplitude profiles of the polariton fringes look smooth at the interface. How to explain such observation?
3. The authors claim that “the hoc overall quality factors remain around 30.... These experimental observations are in agreement with numerical simulations (Supplementary Figure 6 and Note 5).” However, the simulations only show the electric field distribution of the propagating PhP waves, while the quantitative analysis of the hoc factor in the simulations and comparison between simulation and experiment are absent. In this case, I believe it is inadequate to say the experiment are agreement with the simulations.
4. Generally, the tip can also launch PhP and then reflect at a topography abrupt, i.e. the interface. These reflected waves can disturb with the antenna launched waves. The authors are suggested to analysis and clarify this problem.
5. In Figure 3, the experimental near-field optical distribution is different to that in the simulation. For example, in region 2 and 3, 1 and 4, the polaritons fringes are vertically orientated in the s-SNOM image, this observation are very different to that in the simulation. In the region near the defect, the s-SNOM image shows looks like two conical wavefronts. In my opinion, it looks there are some other effects which need to be taken into account and detailly analysed.

Reviewer #2 (Remarks to the Author):

In this manuscript, the authors proposed a method to implement nanoscale polaritonic in-plane cloaking and steering by creating a pseudo-continuous medium. Customized stacked structures are designed and each block contributes layer-specific characteristics that effectively control the flow of polaritons along desired trajectories. It advances the area of transformation optics into the vdW polaritonics domain. The result is interesting, which may ignite the future research on on-chip nanophotonic circuits, I recommend acceptance for publication after revision. My comments and suggestions to the authors are listed below.

1. Generally, the optical cloaks are designed by transformation optics, such as metamaterial cloaks and in-plane graphene plasmon cloak [Science 332, 1291–1294 (2011)]. Here the authors proposed hyperbolic phonon polariton cloak is more like a refractive optics design. What the difference between the two, and how to establish a theoretical framework to design this type of polaritonic cloak.
2. In Fig. 1g, the trend of experimentally measured transmittance changing with the twisted angle doesn't match well with simulated results. Specifically, the measured transmittance even exceeds 100% with $\theta = 90^\circ$.
3. The antenna-launched PhPs propagate from bottom α -MoO₃ film to the 150nm-thick top α -MoO₃ film. The authors provide the quality factor Q and transmittance T to show their high performance during the propagating process. Also, as shown in Fig. 1b to 1f, it seems that the PhPs propagate across the interface of twisted α -MoO₃ regardless the height of the top film. I am wondering if the height of top twisted film affects PhPs propagation, especially for the thickness of several hundred nanometers. Perhaps it is better to give more details about the propagation of PhPs in the interface with thickness difference. Is there any reflection and scattering at the edge of the interface?
4. On the page 9 line 236, "The extracted near-field amplitude profiles demonstrate that the defect has little impact on the intensity and phase of the polariton wave (Figures 3d, e)." However, the phase diagram wasn't plotted in Figures 3d, e.
5. In Fig. 2h, there are background parallel fringes in regions 1-4, the orientations of which vary with the twisted angle. How these fringes excite? It seems extra modes other than antenna-launched PhPs. However, there exists no obvious fringes in Fig. 2f.
6. In Fig. 2h, the propagating hyperbolic PhPs is almost not observed in region 3, whereas signals recovered in region 4. Please comment.
7. On page 4 line 112, the authors claimed that "Besides, the distorted and asymmetrical hyperbolic modes in the twisted region can be interpreted as a shear mode". We know that shear mode origins from off-diagonal terms of dielectric tensor, whereas it doesn't always associate with shear phenomena such as asymmetric distributions of hyperbolic wavefront and dispersion. Please provide more characteristics here to prove it a shear mode.

Reviewer #3 (Remarks to the Author):

H. Teng, N. Chen, and co-authors report the near-field study of hyperbolic phonon-polaritons in composite structures made of biaxial van der Waals crystal α -MoO₃. The authors demonstrate the robust and efficient steering of the propagating polaritons through tailored and assembled two-layer stacks of α -MoO₃ with top and bottom layers twisted relative to each other. This is due to the hybridization between the two layers that results in a mode with controlled topology of the isofrequency contour and the direction of its Poynting vector, depending on the twist angle. I find this work interesting, novel, and timely, and therefore suitable for publication in Nature Communications. However, some revision is necessary. Below I listed my questions and concerns.

Major concerns:

1. Demonstration of polaritonic cloaking is not very convincing: the authors show that the presence of a graphene disk (the “defect”) inside the “cloaked” region is of no consequence for polaritons propagation. However, this alone does not prove that the defect was cloaked unless the “not cloaked” case is demonstrated for comparison. Please show what would happen if polaritons encounter the defect on their way and compare this with the obtained results (Fig. 3 and S11).
2. In relation to the first question, please provide some justification for using the patch of graphene as a defect in the cloaking experiment. I would expect that small area of undoped graphene would have a very limited effect on polaritons in relatively thick α -MoO₃ crystals. Again, demonstration of the “not cloaked” case is necessary.
3. Please elaborate on the reasons behind the high transmittance discussed on page 5. What do the authors mean by “inherent robustness of the topological transition” and “appropriate match of modal profiles”? I suggest to provide a more detailed explanation since the high transmittance at the interfaces is the key phenomenon that allows efficient steering of polaritons. Also, please discuss why the zero twist angle between the α -MoO₃ layers is expected to have the lowest transmittance (highest reflectance) according to the simulations (Fig. S5) – this looks counterintuitive to me.

Other comments:

1. If I understand correctly from Fig. S7, the authors use gold substrate in every experiment, but this is mentioned only once in the context of enhanced near-field signal (line 128). In this case however, observed modes are the “image polaritons” (as in Refs. 39,40) which have different properties compared to polaritons in the same material on a dielectric substrate: different mode profile, significantly shorter wavelength, and longer normalized propagation length which is beneficial for near-field probing. This was demonstrated specifically for α -MoO₃ [Adv. Opt. Mater. 10, 2201492 (2022)] and leveraged in near-field studies of anisotropic plasmons in Ag₂Te [Nat. Mater. 22, 860–866 (2023)] and tunable topological polaritons in graphene/ α -MoO₃ [Nat. Nanotechnol. 17, 940–946 (2022)] in order to obtain better experimental results. Therefore, the gold substrate provides several advantages from the experimental point of view beyond a mere enhancement of the near-field signal. Please discuss the reasons and consequences of using the gold substrate in the manuscript.

2. This work is based on the s-SNOM data, but the explanation of the near-field imaging technique is missing in the manuscript (Supplementary Note 3 provides only technical parameters). Readers not familiar with SNOM may experience difficulty in understanding this work. Please provide concise yet comprehensive explanation of how the near-field imaging of polaritons works: what is imaged in near-field scans, what is the origin of the recorded interference pattern, what is the role of the AFM nano-tip, why using antenna to launch polaritons, etc.

3. It is up to the authors, but I would suggest changing the manuscript's title since the current one may be misleading: the discussion of cloaking takes only three paragraphs in the end of the manuscript, and as the authors themselves noted, the meaning of "cloaking" here is different from the conventional definition. At the same time, from my perspective, the most significant result of this work is the ability to efficiently steer the polaritons while changing their topology in the tailored and assembled multilayers of polaritonic crystals.

4. What is the reason for showing $\text{Re}\{E_z\}$ in the simulation results instead of $|E_z|$ which is measured by SNOM?

5. In Fig. 1a, please indicate the substrate material and clearly indicate that both crystals (top and bottom) are $\alpha\text{-MoO}_3$.

6. Line 113: when mentioning the shear mode, please provide at least a brief explanation of its main characteristics in terms of dispersion since general readers may not be familiar with this new concept.

7. Line 103: Fig. S1 does not show IFCs.

8. Please explicitly state in the text that the near-field amplitude in every figure is normalized so that it is appropriate to compare the different scans and simulations.

9. Line 225: the authors mention an AFM-probe-based transfer method that allows to accurately move the $\alpha\text{-MoO}_3$ microribbons. Please provide some details and/or references regarding this method.

Reviewer 1 (blue text for original comments of the reviewer):

>> In this manuscript, Teng et al constructed interfaces between an in-plane hyperbolic MoO₃ and its twisted double layer, in which in-plane isofrequency curves can be controlled by the twist angle. In this manner, one can control the PhP propagation across the interface and achieve a plenty of noble optical phenomena. Specifically, I believe that planar refraction behavior of hyperbolic polaritons is worthy of study. However, I think the results demonstrated in the current study are still not enough convincing to me due to the following aspects.

We thank the reviewer for the positive evaluation of the basic idea of our work. ("I believe that planar refraction behavior of hyperbolic polaritons is worthy of study."). The reviewer makes constructive suggestions, which have helped us to improve the manuscript, as discussed below.

>>1. When analysis the PhP propagation across the interface between single-layer and double-layer MoO₃, it looks only the waves with momentum along the x direction are considered. How about the PhP with off-axis k ? Did they also refract at the interface and how are their refraction?

We thank the reviewer for raising this important question.

As the reviewer noted, the wave vectors of the off-axis PhPs can also be refracted and adhere to the law of conservation of wave vectors parallel to the interfaces.

For a clearer explanation of refraction with different incident directions, we have shown the IFCs and near-field images of wave vectors incident in several typical different directions (Figure R1). Among them, we marked the refraction process of several typical incident directions. By conserving the polariton wave vector along the direction of the refraction interface, we can obtain the corresponding refraction direction for each incident wave vector. Polariton incidents with different directions undergo distinct angles of refraction. For simplicity, we only label a main direction that corresponds to Poynting vectors of the normally incident wave vector in the manuscript.

Following the reviewer's suggestion, we have added an extra description to explain anisotropic refraction in the revised manuscript (lines 103-109) and updated Supplementary Figures 4m-r, as marked in red:

"We have further marked the refraction process of various typical incident directions. By conserving the polariton wave vector along the direction of the refracting interface, we can obtain the corresponding refraction direction for each incident wave vector (Supplementary Figure 4). Upon transmission of hyperbolic polaritons across the interface, our analysis reveals a transition from normal to negative refraction by rotating the top α -MoO₃ film."

Figure R1. Analysis of polariton refraction for various incident directions. (a, b) Isofrequency contours of hyperbolic polaritons in $\alpha\text{-MoO}_3$ and hybrid polaritons in twisted $\alpha\text{-MoO}_3$. The upper illustrations show a cross-section of each structure. Anisotropic refraction of polaritons takes place at the interface between the two sides of the edge of the upper $\alpha\text{-MoO}_3$ layer due to the conservation of the tangential wave vector components. (c) Experimentally measured near-field amplitude of polaritons illustrating that anisotropic refraction causes a change in the direction of polariton transmission. The twist angle is $\theta = 45^\circ$. The thicknesses of the bottom and top $\alpha\text{-MoO}_3$ films are $t_0=550$ nm and $t_1=150$ nm, respectively. The illumination frequency is 893 cm^{-1} . The scale bar indicates $3 \mu\text{m}$.

Figure R2 (Figures 4m-r of the revised Supplementary Information). (m-r) IFCs of the twisted $\alpha\text{-MoO}_3$ heterostructure for different twist angles, illustrating a transition from normal to negative refraction.

>>2. According to the imaging principle of s-SNOM, a collected near-field amplitude is the sum of the antenna-launched PhPs waves and the local scattering amplitude results from the local dielectric response of the material beneath the tip. Intuitively, the local scattering amplitude at the double-layer should be higher than that at the single-layer, therefore, an amplitude abrupt is expected at the interface between the single- and double-layer region. In Figure S4, the amplitude profiles of the polariton fringes look smooth at the interface. How to explain such observation?

We thank the reviewer for raising this question.

The reviewer is right. An abrupt change in amplitude would be expected at the interface between the single and double-layer regions. For very thin samples, such as graphene on *h*-BN [Ref: Nature Nanotechnology, 2015, 10, 682], the signal intensity indeed varies as a function of the number of layers. However, this abrupt change in amplitude is not found in our experiments; we attribute this observation to the thicker thickness of our α -MoO₃ film and the use of a gold substrate.

As mentioned by the reviewer, the collected near-field amplitude is the combination of antenna-launched PhP waves and the local scattering amplitude resulting from the dielectric response of the material beneath the tip. In our structure, the thickness of the bottom α -MoO₃ is quite substantial, ranging from approximately 300 nm to 550 nm. Consequently, the local scattering amplitude detected by the tip predominantly originates from the material itself. Besides, considering that the tip is already relatively broad, with a size of approximately 25 nm, it is possible that the abruptness may have an impact on a region below the lateral resolution. On the other hand, when dealing with general dielectric substrates like SiO₂, variations in material thickness can lead to changes in the local scattering amplitude, particularly when the thickness is less than half a wavelength. However, the evanescent field is unable to penetrate the metal in our experiments, resulting in total reflection occurring on the gold substrate.

The above explanation aligns with findings from various independent previous studies. We have compiled a set of intensity comparisons illustrating the near-field amplitude of different α -MoO₃ layers when considering variations in material thickness and substrate, as depicted in Figure R3. This observation is consistent with our findings and further supports our explanation.

[REDACTED]

Figure R3. Intensity comparisons from several previous independent studies when considering variations in material thickness.

Following the reviewer's comment, we have added an extra description to explain the near-field signal in the revised Supplementary Information (lines 174-191), as marked in red:

“The collected near-field amplitude is the combination of antenna-launched polaritons and the local scattering amplitude resulting from the dielectric response of the material beneath the tip. An abrupt jump in amplitude would be expected at the interface between the single- and double-layer regions. For very thin samples, such as graphene on *h*-BN,⁵ the intensity signal indeed

varies as a function of the number of layers. However, this abrupt jump in amplitude is not found in our experiments. We attribute this observation to the thicker thickness of our α -MoO₃ film and the use of a gold substrate. In our structure, the thickness of the bottom α -MoO₃ is quite substantial, ranging from approximately 300 nm to 550 nm. Consequently, the local scattering amplitude detected by the tip predominantly originates from the material itself. In addition, when dealing with general dielectric substrates like SiO₂, variations in material thickness can lead to changes in the local scattering amplitude, particularly when the thickness is less than half a wavelength. However, the evanescent field is unable to penetrate the metal in our experiments, resulting in total reflection occurring on the gold substrate. This explanation aligns with findings from various previous independent studies.⁶⁻⁸

>>3. The authors claim that “the *hoc* overall quality factors remain around 30....” These experimental observations are in agreement with numerical simulations (Supplementary Figure 6 and Note 5).” However, the simulations only show the electric field distribution of the propagating PhP waves, while the quantitative analysis of the *hoc* factor in the simulations and comparison between simulation and experiment are absent. In this case, I believe it is inadequate to say the experiment are agreement with the simulations.

We thank the reviewer for this comment. We would like to clarify that the experimental near-field images, including the propagation direction and wavelength of the phonon polaritons, align well with the simulations, rather than focusing on the *hoc* overall quality factors.

To avoid further misunderstandings, we have revised the sentence in the manuscript (lines 144-146), as marked in red: “The experimentally observed propagation direction and fringe wavelength (Figure 1b-f) are in agreement with numerical simulations (Supplementary Figure 9).”

>>4. Generally, the tip can also launch PhP and then reflect at a topography abrupt, i.e. the interface. These reflected waves can disturb with the antenna launched waves. The authors are suggested to analyze and clarify this problem.

We thank the reviewer for raising this important point.

The reviewer is right. In our experiments, there are indeed polariton fringes that are parallel to the edge, which may affect the results. These polariton fringes can be found in our raw data. To address this issue, we have applied a filtering process to effectively eliminate these fringes.

As shown in Figure R4a, we show the raw data, in which one can clearly see the antenna-excited hyperbolic wave, as well as the edge-parallel fringes from both tip- and edge-excited polaritons. To eliminate the disturbance of fringes, we extract antenna-launched phonon polaritons from the complex background signals (Figure R4b) by following the filtering method as described in Ref. [ACS Photonics, 2022, 9, 3696; Nature Nanotechnology, 2022,17, 940].

Figure R4 (Figure 2 of the revised Supplementary Information). Extraction of antenna-launched hybrid polaritons. (a) Raw data of the near-field image, as recorded by our s-SNOM. **(b)** Spurious signals filtered out by our method include tip-launched and edge-launched vertical fringes, among others. **(c)** Near-field image of antenna-launched hybrid polaritons corresponding to the raw data in (a), obtained by subtracting the vertical polariton fringes in (b) from (a). The scale bar indicates 3 μm .

The laser beam under the AFM tip is about $\sim 30 \mu\text{m}$ in lateral size, so it covers a large area of the sample, thus leading to both tip, antenna, and edge launching of polaritons. Therefore, different types of polariton fringes can be identified in our raw data. The black arrows indicate polaritons launched by the antenna. The vertical fringes indicated by the white arrows represent polaritons launched by the $\alpha\text{-MoO}_3$ edges and interference fringes of polaritons launched by the tip and reflected by the edge. The raw data of the near-field image consists of 120×400 pixels (data points),

$$A_0 = \begin{bmatrix} a_1^1 & \cdots & a_1^{400} \\ \vdots & \ddots & \vdots \\ a_{120}^1 & \cdots & a_{120}^{400} \end{bmatrix},$$

where the polaritons launched by the antenna occupy the middle part of the image, while in the area at the edge of the image, only polaritons launched by the tip and the edge do exist, which can be used as background signals,

$$A_1 = \frac{1}{10} \left(\begin{bmatrix} a_1^1 & \cdots & a_1^{400} \\ \vdots & \ddots & \vdots \\ a_5^1 & \cdots & a_5^{400} \end{bmatrix} + \begin{bmatrix} a_{116}^1 & \cdots & a_{116}^{400} \\ \vdots & \ddots & \vdots \\ a_{120}^1 & \cdots & a_{120}^{400} \end{bmatrix} \right).$$

A_1 is obtained from the average value of 10 rows of data from the upper and lower edges to achieve higher accuracy. Therefore, the signal associated with

the polaritons launched by the antenna can be obtained from $B = A_0 - A_1$.

Following the suggestion by the reviewer, we have included Figure R4 as Supplementary Figure 2 and added a more detailed description (lines 156-164) to the revised Supplementary Information to explain the method of extracting antenna-launched phonon polaritons, as marked in red.

>>5. In Figure 3, the experimental near-field optical distribution is different to that in the simulation. For example, in region 2 and 3, 1 and 4, the polaritons fringes are vertically orientated in the s-SNOM image, this observation are very different to that in the simulation. In the region near the defect, the s-SNOM image shows looks like two conical wavefronts. In my opinion, it looks there are some other effects which need to be taken into account and detailly analysed.

We thank the reviewer for raising this point.

We identify that the experimental vertical fringes near the edges (in regions 2, 3, and 1, 4) result from tip and edge excitation. As we collect images through tip scanning mapping, this is an unavoidable interference. However, during simulation, we directly extract the field distribution, eliminating such interference issues, and allowing us to observe the refraction polaritons excited by the antenna. It is important to note that if ribbons 2, 3, or 1, 4 were not vertically split, thereby eliminating the presence of edges, these vertical fringes would not exist. We have effectively demonstrated this scenario in Figure R4.

Regarding the observation that the s-SNOM image appears to show two conical wavefronts in the region near the defect, this can be attributed to fabrication processing errors. In the simulation, all ribbon edges are designed to be perfectly uniform and fit together without any gaps. However, in the experimentally processed sample, the edges of the ribbons exhibit a certain degree of roughness, and there are gaps between them. These rough structures and gap edges further excite and scatter polaritons, leading to undesired spurious signals near the ribbons. However, it is important to note that these effects are primarily localized near the ribbon edges and have minimal impact on the overall results. In future studies, these effects can be mitigated by optimizing and improving the sample fabrication process.

We have added an extra description of the near-field images in the revised Supplementary Information (lines 417-424), as marked in red:

“In regions 2-3 and 1-4, there are some vertically oriented fringes near the vertical edge of each ribbon. This observation is somewhat different from that in the simulations presented in the main text. We identify that these fringes originate in tip and edge excitation. As we collect images through tip-scan mapping, this is an unavoidable interference. However, in the simulation, we directly extract the field distribution, therefore eliminating such interference issues and allowing us to observe the refraction of polaritons excited by the antenna.”

Reviewer 2 (blue text for original comments of the reviewer):

>> In this manuscript, the authors proposed a method to implement nanoscale polaritonic in-plane cloaking and steering by creating a pseudo-continuous medium. Customized stacked structures are designed and each block contributes layer-specific characteristics that effectively control the flow of polaritons along desired trajectories. It advances the area of transformation optics into the vdW polaritonics domain. The result is interesting, which may ignite the future research on on-chip nanophotonic circuits, I recommend acceptance for publication after revision. My comments and suggestions to the authors are listed below.

We thank the reviewer for the very positive comments on our work. (“The result is interesting, which may ignite the future research on on-chip nanophotonic circuits, I recommend acceptance for publication after revision.”). In addition, the reviewer makes constructive and essential suggestions, which have helped us to improve the manuscript, as discussed below.

>>1. Generally, the optical cloaks are designed by transformation optics, such as metamaterial cloaks and in-plane graphene plasmon cloak [Science 332, 1291–1294 (2011)]. Here the authors proposed hyperbolic phonon polariton cloak is more like a refractive optics design. What the difference between the two, and how to establish a theoretical framework to design this type of polaritonic cloak.

We thank the reviewer for raising this point.

We acknowledge that most of the optical cloaking is realized through transformation optics. This approach has revolutionized the field of cloaking with a great deal of freedom, flexibility, and high precision in designing and implementing cloaking devices. However, some optical cloaks employ refractive optics, as shown in Figure R5. They use optical lenses or birefringent crystals to demonstrate cloaking. Transformation optics use spatial variations in refractive index and curved space-time derived from coordinate transformations to achieve arbitrary control of electromagnetic waves. Here we mainly use unconventional refraction of anisotropic materials to control the polariton transmission direction.

We offer nanoscale in-plane cloaking through distinctive refractive behaviors of polaritons by precise stacking and assembling of anisotropic van der Waals materials, thereby creating a spatially varying and adjustable dielectric function along a specific direction.

We would also like to thank the reviewer for raising the intriguing and promising idea that with the combination of transformation optics and hyperbolic polaritons—transformation polaritons, we may conduct further experimental exploration in future studies.

[REDACTED]

Ref: Nature communications, 2011, 2, 176.

Figure R5 (Figure 13 of the revised Supplementary Information). Selection of previous studies on optical cloaking that do not belong to the category of transformation optics. (a) A three-dimensional transmitting and continuously multidirectional cloak using ray optics, albeit with some edge effects¹¹. (b) Invisibility cloak constructed from natural birefringent crystals¹².

Based on the reviewer's suggestion, we have included Figure R5 as Supplementary Figure 13 and added a description to the manuscript to explain the difference between our approach and transformation optics (lines 246-256), as marked in red:

" We note that most of the optical cloaking works in the literature have so far been realized through transformation optics^{53, 54}. This approach has revolutionized the field of cloaking with a great deal of freedom, flexibility, and high precision in designing and implementing cloaking devices. In contrast, our cloaking scheme utilizes anisotropic materials to construct nonuniform media: we employ anisotropic refraction to steer polariton propagation and enable cloaking, which arises from a topological transition in the iso-frequency contours due to the hybridization of different hyperbolic modes. Therefore, our study may provide a foundation for future designs as well as validation of transformative polaritons involving the use of pseudo-continuous media or metamaterials to confine these excitations."

>>2. In Fig. 1g, the trend of experimentally measured transmittance changing with the twisted angle doesn't match well with simulated results. Specifically, the measured transmittance even exceeds 100% with $\theta = 90^\circ$.

We thank the reviewer for raising this question.

We attribute this to the negative refractive effect that produces focusing and enhances transmittance, which causes the measured transmittance to even exceed 100% with $\theta = 90^\circ$.

As the reviewer points out, generally, the transmittance of plane waves for

unidirectional incidence does not exceed 100%. This is due to the partial reflection of energy during the transmission process caused by wave vector mismatch. Regarding the hyperbolic wavefront excited by our point source, different wave vectors result in different angles of refraction. During the transmission through the interface, a phenomenon of refractive focusing occurs, as illustrated in Figure R6. In this case, the amplitude of the wave is the sum of amplitudes with different wave vectors, leading to an increase in amplitude after refraction, resulting in an increase in overall amplitude after refraction. This increase in amplitude can cause the measured transmittance to exceed 100%. Our previous work provides a more detailed description of this refractive focusing effect [Ref: Science 2023, 379, 558; Science Advances, 2021, 7, eabj0127].

Figure R6. Negative refractive produces focusing and enhances the measured transmittance. (a, b) Isofrequency contours of hyperbolic polaritons in $\alpha\text{-MoO}_3$ and hybrid polaritons in twisted $\alpha\text{-MoO}_3$. The upper illustration shows a cross-section of each structure. Refraction of polaritons takes place at the interface between the two sides of the edge of the top $\alpha\text{-MoO}_3$ film due to the conservation of the tangential wave vector components. (c) Experimentally measured near-field amplitude of polaritons illustrating that refraction causes enhanced focusing in the direction of polariton transmission. The twist angle is $\theta = 90^\circ$. The thicknesses of the bottom and top $\alpha\text{-MoO}_3$ films are $t_0=550$ nm and $t_1=150$ nm, respectively. The illumination frequency is fixed at 893 cm^{-1} . The scale bar indicates 3 μm .

Based on the reviewer's comments, we have added a description to the manuscript to clarify this issue (lines 129-132), as marked in red:

“The trend of the change in the experimentally measured transmittance with the twisting angle does not match well with simulations. We attribute this disagreement to the fact that negative refraction produces focusing and enhances the transmittance³⁷.”

>>3. The antenna-launched PhPs propagate from bottom α -MoO₃ film to the 150nm-thick top α -MoO₃ film. The authors provide the quality factor Q and transmittance T to show their high performance during the propagating process. Also, as shown in Fig. 1b to 1f, it seems that the PhPs propagate across the interface of twisted α -MoO₃ regardless the height of the top film. I am wondering if the height of top twisted film affects PhPs propagation, especially for the thickness of several hundred nanometers. Perhaps it is better to give more details about the propagation of PhPs in the interface with thickness difference. Is there any reflection and scattering at the edge of the interface?

We thank the reviewer for raising this question.

The transmittance of polaritons is indeed influenced by the thickness at the top α -MoO₃ layer. The behavior of polariton wave transmission at the interface is primarily determined by two factors: mode field profile matching and wavevector matching.

In terms of wavevector matching, which is more sensitive to thickness variations, we performed calculations to determine the transmittance at a twisting angle $\theta=0^\circ$ for different thicknesses, as depicted in Figure R7. An increase in the thickness of the upper layer results in a larger wavevector mismatch between the single and double layers, leading to a reduction in the overall transmittance. Consequently, when designing a polariton cloaking device, we intentionally select relatively thin top layers to ensure lower losses.

Regarding mode field profile matching, we conducted calculations to determine the cross-sectional profiles of polaritons at different twisting angles, as depicted in Figure R8. Both the single-layer α -MoO₃ and twisted α -MoO₃ exhibit similar mode profiles at the cross-section at various angles. This characteristic ensures that polaritons maintain a relatively high level of transmittance.

Figure R7(Supplementary Figure 8 of the revised Supplementary Information). Numerically simulated polariton transmittance across the interface for different thicknesses of the top $\alpha\text{-MoO}_3$ layer. (a) Simulated near-field ($\text{Re}\{E_z\}$) image illustrating the refractive transmission of polaritons for different thicknesses of the top $\alpha\text{-MoO}_3$ film. (b) Numerically simulated transmittance (red) and wave vector mismatch (blue) as a function of the thickness of the top $\alpha\text{-MoO}_3$ layer.

Figure R8. Numerically simulated refractive transmission of polaritons illustrating mode matching of field profiles for different twisting angles.

We follow the suggestion by the reviewer and include Figure R7 as Supplementary Figure 8 in the revised Supplementary Information to explain the transmittance of polaritons at the interface.

>>4. On the page 9 line 236, “The extracted near-field amplitude profiles demonstrate that the defect has little impact on the intensity and phase of the polariton wave (Figures 3d, e).” However, the phase diagram wasn’t plotted in Figures 3d, e.

We thank the reviewer for pointing out this typo.

We have changed “phase” to “wavelength” and marked it in red in the revised manuscript (line 237).

>>5. In Fig. 2h, there are background parallel fringes in regions 1-4, the orientations of which vary with the twisted angle. How these fringes excite? It seems extra modes other than antenna-launched PhPs. However, there exists no obvious fringes in Fig. 2f.

We thank the reviewer for raising this question.

We attribute these background parallel fringes to polaritons that are excited by the tip and then reflected by impurities on the sample. The impurities are caused by the sample preparation method that we use tip-assisted transfer. The existence of polariton fringes around impurities has been verified in previous studies (Figure R9a) [Ref: Nano Lett., 2017, 17, 5285]. These fringes are only half the wavelength of those excited by the antenna.

In our experiments, a plateau AFM tip was used to act as a bulldozer and push strips into tight splices. Unavoidably, the tip induces physical damage to the ribbons (Figure R9b). The distinct background parallel fringes in regions 1-4 in Figure 2h originate from the polaritons launched by the tip and reflected by these defects, the orientations of which are determined by the IFCs of the twisted area and vary with the twisting angle.

[REDACTED]

[REDACTED]

Figure R9. [REDACTED]

To avoid further misleading, we have removed Figure 2h and updated Figure 2 in the revised Manuscript (lines 147-166) to depict the in-plane steering of polaritons, as illustrated in Figure R10 below.

>>6. In Fig. 2h, the propagating hyperbolic PhPs is almost not observed in region 3, whereas signals recovered in region 4. Please comment.

We thank the reviewer for pointing out this question.

The recovered signals in region 4 originate from polaritons launched by the tip and reflected at the defects of the ribbon, As described in Figure R9 of question 5. The propagation losses of the polariton in Figure 2h accumulate progressively as it traverses through multiple interfaces as a result of the defects in the strip boundaries. Thus, polaritons are only transmitted to region 3 in the experiment and are already considerably weak in intensity.

To avoid further misleading, we have removed Figure 2h and updated **Figure 2 in the revised Manuscript (lines 147-166)** to depict the in-plane steering of polaritons. In the new figure, we have included some control experiments. This includes the normal transmission of polaritons without a microribbon, the introduction of one microribbon to achieve polaritons for changing transmission direction once, and the introduction of two ribbons for changing transmission direction twice. We believe that the new Figure 2 can further clearly demonstrate the polariton steering process, and can also avoid the misunderstandings caused by various spurious signals in previous Figure 2h. The new Figure 2 is shown below:

Figure R10 (Figure 2 of the revised Manuscript). In-plane steering of polaritons. (a) Illustration of tailored α -MoO₃ microribbons with different cut angles β relative to the crystallographic orientation of a common source film. (b-g) Optical images (b, d, f) and atomic force microscopy (AFM) images (c, e, g) of different polaritonic devices composed of a bottom α -MoO₃ film and tailored α -MoO₃ microribbons. Two α -MoO₃ microribbons with $\beta=45^\circ$ (labeled 1) and $\beta=135^\circ$ (labeled 2) are used in panels (b-g). (h, j, l) Near-field amplitude images corresponding to the devices in panels (b, d, f), respectively. The polariton propagation path is controlled by the top tailored α -MoO₃ microribbons: one deflection at the microribbon labeled 1 in (j), leading to a lateral shift of polaritons; and two deflections with opposite angles in (l), leading to a final undeflected transmitted beam. Red-dashed arrows indicate the polariton propagation direction dictated by the Poynting vector \mathbf{S} , as obtained from the IFCs analysis presented in panels (i, k, m). The thicknesses of the bottom film and top ribbons are $t_0=184$ nm and $t_1=154$ nm in (h, j, l). Scale bars in panels (b, d, f, h, j, l) indicate 3 μm . (i, k, m) Calculated polariton IFCs (blue curves) corresponding to each region in the devices. Horizontal black lines (labeled ①-④) indicate interfaces between different regions. Red arrows represent Poynting vectors \mathbf{S} , directed along the energy flow and normal to the IFCs. Scale bars indicate $20 k_0$.

>>7. On page 4 line 112, the authors claimed that “Besides, the distorted and asymmetrical hyperbolic modes in the twisted region can be interpreted as a shear mode”. We know that shear mode originates from off-diagonal terms of the dielectric tensor, whereas it doesn't always associate with shear phenomena such as asymmetric distributions of hyperbolic wavefront and dispersion. Please provide more characteristics here to prove it a shear mode.

We thank the reviewer for this comment.

The reviewer is right. Shear mode arises from the inability to diagonalize the dielectric tensor space. It has been found in low-symmetry monoclinic and triclinic crystals [Ref: Nature 2022, 602, 595; Nature Nanotechnology, 2023, 18, 64]. Similarly, the dielectric function in twisted α -MoO₃ can be expressed as a linear superposition of the dielectric functions of multiple monolayers α -MoO₃ [Ref: Nature Materials, 2023, 22,867; Optics Lett., 2022, 47, 5433].

$$\epsilon_{total} = \sum_{j=1}^n \epsilon_{MoO_3}^j = \begin{bmatrix} \sum_{j=1}^n \epsilon_x^j (\cos^2(\theta_j) + \sin^2(\theta_j)) & \sum_{j=1}^n \epsilon_x^j \epsilon_y^j \cos(\theta_j) \sin(\theta_j) & 0 \\ \sum_{j=1}^n \epsilon_x^j \epsilon_y^j \cos(\theta_j) \sin(\theta_j) & \sum_{j=1}^n \epsilon_y^j (\sin^2(\theta_j) + \cos^2(\theta_j)) & 0 \\ 0 & 0 & \sum_{j=1}^n \epsilon_z^j \eta_j \end{bmatrix}, (S1)$$

where η_j denotes the ratio of the thickness of each layer to the total thickness.

From the dielectric function of twisted α -MoO₃, it can be seen that there are non-diagonal terms due to the introduction of the twisting angle θ resulting in the non-parallel optical axes on different layers. As a result, the distorted and asymmetrical hyperbolic modes in the twisted region of α -MoO₃ can be characterized as shear modes.

Following the reviewer's comment, we have included a further explanation of the shear mode in the revised manuscript (lines 116-121), as marked in red:

“Besides, the dielectric function in twisted α -MoO₃ can be expressed as a linear superposition of the dielectric functions of the layers⁴², which results in the presence of off-diagonal terms. Therefore, the distorted and asymmetrical hyperbolic modes in the twisted region can be interpreted as a shear mode that has recently been observed in natural crystals with low symmetry such as monoclinic β -Ga₂O₃ and CdWO₄^{24, 25}.”

Reviewer 3 (blue text for original comments of the reviewer):

>> H. Teng, N. Chen, and co-authors report the near-field study of hyperbolic phonon-polaritons in composite structures made of biaxial van der Waals crystal α -MoO₃. The authors demonstrate the robust and efficient steering of the propagating polaritons through tailored and assembled two-layer stacks of α -MoO₃ with top and bottom layers twisted relative to each other. This is due to the hybridization between the two layers that results in a mode with controlled topology of the isofrequency contour and the direction of its Poynting vector, depending on the twist angle. I find this work interesting, novel, and timely, and therefore suitable for publication in Nature Communications. However, some revision is necessary. Below I listed my questions and concerns.

We thank the reviewer for the very positive comment on our work. (“I find this work interesting, novel, and timely, and therefore suitable for publication in Nature Communications.”). The reviewer makes constructive suggestions, which have helped us to improve the manuscript, as discussed below.

>>1. Demonstration of polaritonic cloaking is not very convincing: the authors show that the presence of a graphene disk (the “defect”) inside the “cloaked” region is of no consequence for polaritons propagation. However, this alone does not prove that the defect was cloaked unless the “not cloaked” case is demonstrated for comparison. Please show what would happen if polaritons encounter the defect on their way and compare this with the obtained results (Fig. 3 and S11).

We thank the reviewer for this comment.

We have added a new experiment to address this issue. When the defect (a graphite disk) is introduced in the propagation path of polaritons, it hinders the transmission of polaritons. More specifically, the scattering and reflection in the defect edge result in a reduction in polaritonic intensity (Figure R11).

Figure R11 (Supplementary Figure 16 of the revised Supplementary Information). Control experiments for comparison of polariton propagation with and without introducing a defect when there is no cloaking structure. (a, b) Near-field amplitude images of polariton propagation before and after introducing a defect when there is no cloaking structure. (c) Analysis of defect-induced polariton fringes by extracting near-field profiles

from panels (a) and (b). When there are no defects, the hyperbolic wave excited by the antenna can be transmitted normally. After placing a defect in the path of the polaritons, the fringes almost disappear in the space behind the defect.

Following the comment by the reviewer, we have included a new section in the revised Supplementary Information (lines 426-435) and added a description in the revised manuscript (lines 240-245) to clarify how a defect influences polariton transmission in the absence of cloaking structure, as marked in red:

“In contrast, the presence of a defect along the propagation path of the polaritons without a cloaking structure induces a substantial reduction in the intensity (Supplementary Figure 16). This effect is primarily attributed to the scattering and reflection of propagating polaritons at the defect edge as well as the modified dielectric environment within the defect region.”

>>2. In relation to the first question, please provide some justification for using the patch of graphene as a defect in the cloaking experiment. I would expect that small area of undoped graphene would have a very limited effect on polaritons in relatively thick α -MoO₃ crystals. Again, demonstration of the “not cloaked” case is necessary.

We thank the reviewer for raising this point.

The reviewer is right. A small area of undoped monolayer graphene disk would have virtually no effect on the propagation of phonon polaritons in a relatively thick α -MoO₃. However, we need to clarify that we employed a graphite disk (50 nm-thickness and 1 μ m diameter) instead of a graphene.

We chose the graphite disk as the defect for the following three reasons: Firstly, graphite is easy to process and transfer, allowing us to obtain samples with precise structures. Secondly, it can interfere with the transmission of polaritons through processes such as reflection and scattering. Lastly, graphite does not excite excessive polaritons like a gold disk, thus minimizing interference with experimental observations.

Following the comment by the reviewer, we have added a description in the revised Supplementary Information (lines 104-109) to explain the reason we chose a graphite disk as the defect.

“In the cloaking device, we also used a graphite disk with 50 nm thickness and 1 μ m diameter as a defect for the following reasons: firstly, this structure is easy to process and allows for precise manipulation; secondly, it introduces significant interference in the transmission of polaritons; and thirdly, it avoids excessive excitation of polaritons, minimizing interference with experimental observations.”

>>3. Please elaborate on the reasons behind the high transmittance discussed on page 5. What do the authors mean by “inherent robustness of the topological transition” and “appropriate match of modal profiles”? I suggest to provide a more detailed explanation since the high transmittance at the interfaces is the key phenomenon that allows efficient steering of polaritons. Also, please discuss why the zero twist angle between the α -MoO₃ layers is expected to have the lowest transmittance (highest reflectance) according to the simulations (Fig. S5) – this looks counterintuitive to me.

We thank the reviewer for raising these questions. To provide a comprehensive response, we divide it into two parts:

3.1 Discussion of high transmittance at the interface

To achieve high transmittance at the interface, wavevector matching and mode field profile matching need to be considered. For the different twisted structures, we have analyzed these two conditions through numerical simulation, as shown in Figure R12.

In terms of mode field profile matching, the near-field ($\text{Re}\{E_z\}$) distribution of polaritons on both sides of the interface where refraction occurs is uniform (Figure R12a), and there is no strong distortion induced by different structures and modes on both sides. This is because the polaritons coupling between the double-layer α -MoO₃ has a robust nature, which means the hybrid polaritons are not affected by continuous small perturbations or disorder [Ref: Nature, 2020, 582, 209.]

We further extracted the transmittance corresponding to the degree of wave vector mismatching. We selected one of the most representative normal incident wave vectors in IFC and analyzed its transmission in a single layer and its refraction at different twisted structures. The relative wave vector mismatch is maintained at less than 20%. Especially when the rotation angle is between 20 and 60 degrees, the mismatch rate is less than 3%.

To sum up, our structure meets high mode field distribution matching and wave vector matching, so it shows high transmittance in the measurement.

We follow the suggestion by the reviewer and include Figure R7 as Supplementary Figure 8 in the revised Supplementary Information to explain the transmittance of polaritons at the interface.

3.2 Discussion of the lowest transmittance of zero-twist angle structure

As shown in Figure R12b, the wave vector mismatch is maximized at the twisting angles $\theta=0^\circ$ and 90° . Thereby one can expect a lower transmittance of twisting angle $\theta=0^\circ$ when compared with other angles. Note that, why $\theta=90^\circ$ is different from $\theta=0^\circ$ is because the negative refractive effect produces focusing and enhances measured transmittance. This has been detailed and discussed in Figure R5 of this response letter.

Figure R12. Numerically simulated spatial distribution and transmission of polaritons as they reach the twisted region for different twisting angles. (a) Numerically simulated spatial distributions ($\text{Re}\{E_z\}$) of polariton transmission across the interface for various twisting angles θ . Polaritons are launched from a port to the left and then propagate in the bottom $\alpha\text{-MoO}_3$ film to the twisted region toward a port to the right. (b) Numerically simulated wave vector mismatch as a function of twisting angle θ .

Other comments:

>>1. If I understand correctly from Fig. S7, the authors use gold substrate in every experiment, but this is mentioned only once in the context of enhanced near-field signal (line 128). In this case however, observed modes are the “image polaritons” (as in Refs. 39,40) which have different properties compared to polaritons in the same material on a dielectric substrate: different mode profile, significantly shorter wavelength, and longer normalized propagation length which is beneficial for near-field probing. This was demonstrated specifically for $\alpha\text{-MoO}_3$ [Adv. Opt. Mater. 10, 2201492 (2022)] and leveraged in near-field studies of anisotropic plasmons in Ag_2Te [Nat. Mater. 22, 860–866 (2023)] and tunable topological polaritons in graphene/ $\alpha\text{-MoO}_3$ [Nat. Nanotechnol. 17, 940–946 (2022)] in order to obtain better experimental results. Therefore, the gold substrate provides several advantages from the experimental point of view beyond a mere enhancement of the near-field signal. Please discuss the reasons and consequences of using the gold substrate in the manuscript.

We thank the reviewer for this comment as well as for pointing out these relevant works.

Following the reviewer's comments, we have added these four pioneering relevant works [Nature Communications, 2020, 11, 3649; Advanced Optical Materials, 2022, 10, 2201492; Science Advances, 2022, 8, eabn0627; Nature Materials 2023, 22, 860] as new citations. We have also provided detailed explanations in the revised manuscript to explain the advantages of using a metallic substrate, (lines 136-141), as marked in red.

“Besides, gold flakes serve as a flat low-loss substrate in our experiments because a new image mode is formed, which stems from the coupling between collective charge oscillations and hybridization of polaritons with their mirror image in the metal. Notably, the image phonon polaritons provide both stronger field confinement and a longer lifetime compared to phonon polaritons on a dielectric substrate.⁴³⁻⁴⁷”

>>2. This work is based on the s-SNOM data, but the explanation of the near-field imaging technique is missing in the manuscript (Supplementary Note 3 provides only technical parameters). Readers not familiar with SNOM may experience difficulty in understanding this work. Please provide concise yet comprehensive explanation of how the near-field imaging of polaritons works: what is imaged in near-field scans, what is the origin of the recorded interference pattern, what is the role of the AFM nano-tip, why using antenna to launch polaritons, etc.

We thank the reviewer for this suggestion.

Following the reviewer's suggestion, we have added a comprehensive explanation of the near-field imaging technique in the revised Supplementary Note 2 (lines 110-164) and included Figure R13 as Supplementary Figure 1 in the revised Supplementary Information (lines 165-168), as marked in red.

“**Note 2. Scattering-type scanning near-field optical microscopy (s-SNOM) measurements**”

2.1 s-SNOM measurements

We utilized a commercially available s-SNOM (Neaspec GmbH) to perform infrared nanoimaging of polaritons in α -MoO₃. The system employed a platinum-coated atomic force microscope tip (NanoWorld) with an approximate radius of 25 nm as the primary scanning platform for approaching and scanning the sample. A monochromatic mid-infrared light source from a quantum cascade laser (QCL) with a tunable frequency range of 890 to 2000 cm⁻¹ was used to illuminate the tip. The laser beam, with p-polarization and a lateral spot size of around 25 μ m, was focused through a parabolic mirror at an incident angle of 55° to 65°. This setup effectively covered a large area of interest in the samples. The near-field nanoimages were captured by a pseudoheterodyne interferometric detection module, with the AFM tip-tapping frequency and amplitudes set to approximately 270 kHz and 30-50 nm, respectively. The detected signal was demodulated at the third harmonic (denoted **S₃**) of the

tapping frequency to obtain near-field amplitude images that were free of any background interference.

2.2 Polariton launched by gold antenna

In s-SNOM measurements, polaritons can be excited by a variety of structures, including tips, antennas, edges, and even defects. We primarily utilize metal antennas as the excitation source due to their ability to separate the excitation and detection processes. This separation enables us to directly observe a diversity of refractive transmission of polaritons. When tip excitation is employed, only the interference fringes of the mode would be observed, and the refractive transmission cannot be directly visualized.

In addition, metal antennas can provide high excitation efficiency. When infrared light irradiates the metal antenna, it can excite the plasmon resonance in the antenna and form an in-plane oscillating dipole, thereby exciting the polaritons in the sample with high efficiency. To do this, we designed resonant antennas with a length of approximately 3.0 μm .

2.3 Discussion of fringe formation

The signals observed in near-field imaging can be categorized into five pathways of light in the measurement, as depicted in Supplementary Figure 1. These paths are classified as P_1 to P_5 . P_1 denotes the fraction of incident light that is directly reflected back by the tip (black, tip-reflected). P_2 shows the outward propagation of polaritons launched by the tip, which are subsequently scattered into the detector by the sample edge (blue, tip-launched/edge-scattered). P_3 involves tip-launched polaritons that propagate outward to the edge, reflect, and then scatter into the detector after interacting with the tip (orange, tip-launched/edge-reflected/tip-scattered). P_4 illustrates polaritons launched from the edge and scattered into the detector by the tip (green, edge-launched/tip-scattered). Lastly, P_5 demonstrates resonant antenna-launched polaritons that are scattered into the detector by the tip (red, antenna-launched/tip-scattered). More details about the discussion and analysis of fringe formation have been reported in several previous studies^{1,2}.

In s-SNOM measurements, various structures such as tips, antennas, edges, and even defects can serve as exciters for polaritons. Although we primarily utilize metal antennas as the excitation source, one can expect that the antenna-launched hyperbolic wave should be surrounded by edge-parallel fringes from both tip- and edge-launched polaritons, as illustrated in Supplementary Figure 2a. To eliminate the disturbance of fringes, we extract antenna-launched phonon polaritons from the complex background signals (Supplementary Figure 2b) by following a filtering method described in previous studies^{3,4}.

Figure R13 (Figure 1 of the revised Supplementary Information) Illustration of different paths from which the signals can be collected by the detectors. P₁ to P₅ represent different signal pathways collected by the s-SNOM detectors.

>>3. It is up to the authors, but I would suggest changing the manuscript's title since the current one may be misleading: the discussion of cloaking takes only three paragraphs in the end of the manuscript, and as the authors themselves noted, the meaning of "cloaking" here is different from the conventional definition. At the same time, from my perspective, the most significant result of this work is the ability to efficiently steer the polaritons while changing their topology in the tailored and assembled multilayers of polaritonic crystals.

We thank the reviewer for this suggestion.

We agree with the reviewer and we have revised the title to "**Steering and cloaking of hyperbolic polaritons at deep-subwavelength scales.**" This title can encompass the entire research content of this work.

>>4. What is the reason for showing $\text{Re}\{E_z\}$ in the simulation results instead of $|E_z|$ which is measured by SNOM?

We thank the reviewer for raising this question.

Demodulating the s-SNOM signal at higher harmonic frequencies of the tapping frequency can generate real near-field signals and can be summarized by the following two equations:

$$S_n(\omega) = \int_0^T e^{in\Omega t} E_{rad}(\omega, \Omega, t) dt, \quad (S2)$$

$$d = d_0 + A[1 + \sin(\Omega T)]. \quad (S3)$$

where S_n represents the amplitude signal at the n^{th} harmonic, Ω is the oscillation frequency of the tip, d is the distance between the tip and the sample. E_{rad} represents the scattered signal from the tip, which carries information about the sample. In experimental measurements, this scattered signal E_{rad} is complex. As a simplification, treating the tip as an electric dipole is an effective model. Besides, we neglect the horizontal dipole moments of the tip due to the slender shape of the tip, which is weaker by several orders of magnitude than the polarization in the z -direction. At this point, we can approximately consider the experimental scattered field E_{rad} to be proportional to the electric field E_z generated by the electric dipole. Therefore, in the process of using COMSOL simulation, we only extract the E_z -component of the electric field to describe the propagation of polaritons. This processing approach also has been reported in previous studies [Ref: Science, 2014, 344, 1369; Nano Lett. 2011, 11, 4701; Phys. Rev B. 2014, 90, 085136].

>>5. In Fig. 1a, please indicate the substrate material and clearly indicate that both crystals (top and bottom) are $\alpha\text{-MoO}_3$.

We have followed the suggestion by the reviewer and the new Figure 1 is shown below.

Figure R14 (Figure 1 of the revised manuscript).

>>6. Line 113: when mentioning the shear mode, please provide at least a brief explanation of its main characteristics in terms of dispersion since general readers may not be familiar with this new concept.

We thank the reviewer for this suggestion.

According to the reviewer's comment, we have included a further explanation of the shear mode in the revised manuscript (lines 116-121), as marked in red:

“Besides, the dielectric function in twisted α -MoO₃ can be expressed as a linear superposition of the dielectric functions of the layers⁴², which results in the presence of off-diagonal terms. Therefore, the distorted and asymmetrical hyperbolic modes in the twisted region can be interpreted as a shear mode that has recently been observed in natural crystals with low symmetry such as monoclinic β -Ga₂O₃ and CdWO₄^{24, 25}”

>> 7. Line 103: Fig. S1 does not show IFCs.

We thank the reviewer for pointing out this error.

We have revised the description in lines 101-103, as marked in red:

“see the theoretical model in Supplementary Figure 3 and Note 3 along with the corresponding isofrequency contours (IFCs) in Supplementary Figure 4”

>>8. Please explicitly state in the text that the near-field amplitude in every figure is normalized so that it is appropriate to compare the different scans and simulations.

We have added a description of the normalized near-field amplitude in the revised manuscript (lines 81-82), as marked in red:

“Note that experimental near-field images are normalized in this work, as well as the simulated images.”

We thank all reviewers again for their valuable comments.

Yours sincerely,

The authors

REVIEWERS' COMMENTS

Reviewer #1 (Remarks to the Author):

The authors have addressed my previous concerns on their results. I suggest that the manuscript to be accepted in its current form.

Reviewer #2 (Remarks to the Author):

The authors have successfully addressed my comments, thus I recommend it for publication in its current version.

Reviewer #3 (Remarks to the Author):

Authors thoroughly addressed all comments. Thus I recommend the manuscript for publication in its current form.